# Reactions of Cadmium(II) Halides and Di-2-Pyridyl Ketone Oxime: One-Dimensional Coordination Polymers

**DOI:** 10.3390/molecules29020509

**Published:** 2024-01-19

**Authors:** Christina Stamou, Pierre Dechambenoit, Zoi G. Lada, Patroula Gkolfi, Vassiliki Riga, Catherine P. Raptopoulou, Vassilis Psycharis, Konstantis F. Konidaris, Christos T. Chasapis, Spyros P. Perlepes

**Affiliations:** 1Department of Chemistry, University of Patras, 26504 Patras, Greece; xrstamou@gmail.com (C.S.); patroula.gkolfi@gmail.com (P.G.); rigavaso@gmail.com (V.R.); 2Centre de Recherche Paul Pascal, UMR 5031, CNRS, University of Bordeaux, 33600 Pessac, France; dechambenoit@crpp-bordeaux.cnrs.fr; 3Institute of Chemical Engineering Sciences, Foundation for Research and Technology-Hellas (FORTH/ICE-HT), Platani, P.O. Box 1414, 26504 Patras, Greece; zoilada@iceht.forth.gr; 4Institute of Nanoscience and Nanotechnology, NCSR “Demokritos”, 15310 Aghia Paraskevi Attikis, Greece; c.raptopoulou@inn.demokritos.gr; 5Department of Chemistry, Materials Science and Chemical Engineering “Giulio Natta”, Via L. Mancinelli 7, 20131 Milan, Italy; 6Institute of Chemical Biology, National Hellenic Research Foundation, 11635 Athens, Greece

**Keywords:** cadmium(II) complexes, coordination chemistry, di-2-pyridyl ketone oxime, spectroscopic (IR, Raman, and ^1^H, ^13^C, and ^113^Cd NMR) studies, structural studies

## Abstract

The coordination chemistry of 2-pyridyl ketoximes continues to attract the interest of many inorganic chemistry groups around the world for a variety of reasons. Cadmium(II) complexes of such ligands have provided models of solvent extraction of this toxic metal ion from aqueous environments using 2-pyridyl ketoxime extractants. Di-2-pyridyl ketone oxime (dpkoxH) is a unique member of this family of ligands because its substituent on the oxime carbon bears another potential donor site, i.e., a second 2-pyridyl group. The goal of this study was to investigate the reactions of cadmium(II) halides and dpkoxH in order to assess the structural role (if any) of the halogeno ligand and compare the products with their zinc(II) analogs. The synthetic studies provided access to complexes {[CdCl_2_(dpkoxH)∙2H_2_O]}_n_ (**1**∙2H_2_O), {[CdBr_2_(dpkoxH)]}_n_ (**2**) and {[CdI_2_(dpkoxH)]}_n_ (**3**) in 50–60% yields. The structures of the complexes were determined by single-crystal X-ray crystallography. The compounds consist of structurally similar 1D zigzag chains, but only **2** and **3** are strictly isomorphous. Neighboring Cd^II^ atoms are alternately doubly bridged by halogeno and dpkoxH ligands, the latter adopting the η^1^:η^1^:η^1^:μ (or 2.0111 using Harris notation) coordination mode. A terminal halogeno group completes distorted octahedral coordination at each metal ion, and the coordination sphere of the Cd^II^ atoms is {Cd^II^(η^1^ − X)(μ − X)_2_(N_pyridyl_)_2_(N_oxime_)} (X = Cl, Br, I). The *trans*-donor–atom pairs in **1**∙2H_2_O are Cl_terminal_/N_oxime_ and two Cl_bridging_/N_pyridyl_; on the contrary, these donor–atom pairs are X_terminal_/N_pyridyl_, X_bridging_/N_oxime,_ and X_bridging_/N_pyridyl_ (X = Br, I). There are intrachain H-bonding interactions in the structures. The packing of the chains in **1**∙2H_2_O is achieved via π-π stacking interactions, while the 3D architecture of the isomorphous **2** and **3** is built via C-H∙∙∙C_g_ (C_g_ is the centroid of one pyridyl ring) and π-π overlaps. The molecular structures of **1**∙2H_2_O and **2** are different compared with their [ZnX_2_(dpkoxH)] (X = Cl, Br) analogs. The polymeric compounds were characterized by IR and Raman spectroscopies in the solid state, and the data were interpreted in terms of the known molecular structures. The solid-state structures of the complexes are not retained in DMSO, as proven via NMR (^1^H, ^13^C, and ^113^Cd NMR) spectroscopy and molar conductivity data. The complexes completely release the coordinated dpkoxH molecule, and the dominant species in solution seem to be [Cd(DMSO)_6_]^2+^ in the case of the chloro and bromo complexes and [CdI_2_(DMSO)_4_].

## 1. Introduction

The oxime group (
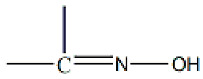
), which is commonly produced by the reaction of a carbonyl group (from an aldehyde giving aldoximes and from a ketone resulting in ketoximes) with hydroxylamine, is a classical functional group in organic chemistry. It also plays a significant role in supramolecular chemistry and crystal engineering due to its capability to form several types of H bonds. The landmark of the use of oximes was the gravimetric determination of Ni(II) as the highly insoluble, red bis(dimethylglyoximato)nickel(II) solid [1]. Oxime and oximato metal complexes are central “players” in several aspects of coordination [2,3] and bioinorganic [4,5] chemistry, molecular magnetism [6,7], and homogeneous catalysis [8]. The reactivity of the coordinated oxime group is also of interest; this group has three sites (C, N, and O) for electrophilic or nucleophilic additions, most often promoted/assisted by metal ions [9,10,11]. Such reactions can proceed with rupture or preservation of the {CNO} moiety. In many cases, the oxime group is part of an organic molecule that contains one or more donor sites. Typical examples are the various 2-pyridyl oximes (Figure 1, left; R = a non-donor group). These ligands have been used among other purposes, including the following: (i) the linking (by coordination bonds) of smaller clusters to supramolecular entities [12]; (ii) the synthesis of single-chain magnets (SCMs) [13]; (iii) the isolation of 3d/4f-metal dinuclear and polynuclear complexes [14]; (iv) the “switching on” of single-molecule magnetism (SMM) properties [15]; and (v) the modeling of solvent extraction of Cd(II) from aqueous media using 2-pyridyl ketoxime extractants [16].

Di-2-pyridyl ketone oxime (Figure 1, right, abbreviated as dpkoxH, where H denotes the potentially acidic hydrogen atom) occupies a special position among the 2-pyridyl oximes. Not only is the R group a donor site but also this group is a second 2-pyridyl unit. An attractive aspect of dpkoxH (and its anionic form, dpkox^−^) is its remarkable coordinative flexibility and versatility that have resulted in 3d- and 3d/4f-metal coordination clusters with interesting structural features and magnetic properties [17,18]. Another area to which the deprotonated ligand is relevant is the chemistry of metallacrowns (MCs), both regular and inverse [19,20]. Finally, an equally interesting aspect of the dpkoxH coordination chemistry is the activation of this molecule by 3d-metal ions, which appears to be useful in synthetic inorganic chemistry [21].

Cadmium(II)/pyridyl *ald*oxime complexes are known in the literature. This chemistry has been pioneered mainly by Forani’s group, and her work has led to complexes with remarkable structures and impressive properties, including luminescence and supramolecular isomerism [22,23,24,25,26].

One family of such Cd(II) compounds includes mixed carboxylate/pyridyl aldoxime complexes. The complexes {[Cd(1,3-bdc)(2-pyaoH)]}_n_ and {[Cd(1,4-bdc)(4-pyaoH)_2_]∙DMF}_n_ have laminar 2D structures, whereas {[Cd(fum)(2-pyaoH)_2_]}_n,_ {[Cd(1,3-bdc)(4-pyaoH)(H_2_O)_2_]∙DMF∙H_2_O}_n_ and {[Cd(1,4-bdc)(4-pyaoH)_2_(H_2_O)]∙DMF}_n_ are 1D polymers; in these formulae, 2-pyaoH is pyridine-2-aldoxime, 4-pyaoH is pyridine-4-aldoxime, fum^2−^ is the fumarate(-2) ligand, 1,3-bdc^2−^ is the 1,3-benzenedicarboxylate(-2) ligand, and 1,4-bdc^2−^ is the 1,4-benzenedicarboxylate(-2) ligand [22]. The complexes [Cd_2_(O_2_CMe)_4_(4-pyaoH)_4_]∙4H_2_O and [Cd(O_2_CMe)_2_(4-pyaoH)_3_]∙3H_2_O consist of discrete (dinuclear and mononuclear, respectively) molecules and display blue luminescence upon excitation with UV light [23]. The use of the sulfato co-ligand in Cd(II)/4-pyaoH chemistry has led to the 1D compound {[Cd(SO_4_)(4-pyaoH)_2_(H_2_O)_2_]}_n,_ which displays ligand-based emission [24]; this complex is a supramolecular isomer with its Zn(II) counterpart. The complexes [Cd_2_(suc)(2-pyaoH)_4_(H_2_O)_2_](BF_4_)_2_, {[Cd(suc)(2-pyaoH)_2_]}_n_, {[Cd(mal)(4-pyaoH)(H_2_O)]}_n,_ and {[Cd(adi)(4-pyaoH)_2_]∙DMF}_n_ were isolated using the saturated aliphatic dicarboxylate(-2) ligands malonate (mal^2−^), succinate (suc^2−^), and adipate (adi^2−^). The polymeric succinate complex has a 1D array, whereas the other two coordination polymers have layered 2D structures [25]. Further use of the 2-pyaoH ligand in Cd(II) chemistry has resulted in the isolation of the mononuclear complex [Cd(HCO_2_)_2_(2-pyaoH)_2_], and the 1D coordination polymers {[Cd(1,4-bdc)(2-pyaoH)]∙1.5DMF}_n_ and {[Cd(SO_4_)(2-pyaoH)(H_2_O)]}_n_; a rare dual light emission was observed in the sulfato compound which, according to TD/DFT calculations, originates from the nπ* (at 400 nm) and ππ* (at 650 nm) of the 2-pyaoH ligand [26]. The complex [Cd(O_2_CMe)_2_(2-pyaoH)_2_] [27] has a similar structure to that of [Cd(HClO_2_)_2_(2-pyaoH)_2_] [26], the two monodentate carboxylate ligands being in *cis* positions. Our group has also contributed to the Cd(II)/pyridyl *ald*oxime chemistry by isolating and characterizing the mononuclear complex [CdI_2_(2-pyaoH)_2_] and the 1D chain compounds {[CdI_2_(3-pyaoH)_2_]}_n_ and {[CdI_2_(4-pyaoH)_2_]}_n_ [16], where 3-pyaoH is pyridyl-3-aldoxime; the 2-pyaoH molecules behave as N_pyridyl_ and N_oxime_-bidentate chelating ligands, whereas 3-pyaoH and 4-pyaoH act in a monodentate manner coordinating via their pyridyl nitrogen atom. 

The Cd(II)/pyridyl *ket*oxime chemistry is much less studied, and almost all of the published work comes from our groups. The reaction between CdSO_4_∙8/3H_2_O and methyl 2-pyridyl ketone oxime (mepaoH, R = Me in Figure 1) in H_2_O gives {[Cd(SO_4_)(mepaoH)(H_2_O)]}_n_∙{[Cd(SO_4_)(mepaoH)(H_2_O)_2_]}_n_, whose unique structure consists of two different linear and ladder-type units [28]. The mononuclear complexes [Cd(phpaoH)_3_](NO_3_)_2_ [29], [CdCl_2_(phpaoH)_2_]∙H_2_O [30], and the 1D coordination polymer {[CdCl_2_(phpaoH)]}_n_ [30], where phpaoH is phenyl 2-pyridyl ketone oxime (R = Ph in Figure 1), have also been prepared and structurally characterized. The mepaoH and phpaoH ligands chelate Cd(II) via their N_pyridyl_ and N_oxime_ atoms.

However, Cd(II)/dpkoxH complexes remain unknown in the literature. In this work, we are glad to report the first Cd(II) complexes with this ligand by describing the syntheses, structures, and spectroscopic characterization of the products derived from the reactions of cadmium(II) halides and dpkoxH; an attempt has also been made to study the behavior of the compounds in solution. 

## 2. Results and Discussion

### 2.1. Synthetic Comments

For reasons outlined in the Introduction section, we sought the preparation of Cd(II)/dpkoxH complexes. The best results (from the crystallization viewpoint) were obtained using cadmium(II) halides. We started our efforts using neutral dpkoxH, i.e., by avoiding adding an external base in the reaction mixtures, which would lead to the deprotonation of the oxime group. Reactions with the addition of bases are in progress in our laboratories, and—if successful—they will be submitted in the future. A variety of CdX_2_/dpkoxH (X = Cl, Br, and I) reaction systems using different reagent molar ratios, solvent media, temperatures, and crystallization techniques were systematically employed before arriving at the optimized preparative conditions described in Section 3. Equations (1)–(3) illustrate the synthesis of the complexes.
(1)$$\begin{array}{c} n\;{\mathrm{CdCl}}_{2}\cdot 2H_{2}O+n\;\mathrm{dpkoxH}\;\overset{\mathrm{MeOH}}{\to }\;\{[{\mathrm{CdCl}}_{2}(\mathrm{dpkoxH})]\cdot 2H_{2}O{\}}_{n}\\ \;\;\;\;\;\;\;\;\;\;\;\;\;\;\;\;\;\;\;\;\;\;\;\;\;\;\;\;\;\;\;\;\;\;\;\;\;\;\;\;\;\;\;\;\;\;\;\;\;\;\;\;\;\;\;\;\;\;1\cdot 2H_{2}O \end{array}$$
(2)$$\begin{array}{c} n\;{\mathrm{CdBr}}_{2}\cdot 4H_{2}O+n\;\mathrm{dpkoxH}\;\overset{\mathrm{MeCN}}{\to }\;\{[{\mathrm{CdBr}}_{2}(\mathrm{dpkoxH})]{\}}_{n}+4n\;H_{2}O\\ \;\;\;\;\;\;\;\;\;\;\;\;\;\;\;\;\;\;\;\;\;\;\;\;\;\;\;\;\;\;\;\;\;\;\;\;\;\;\;\;\;\;\;\;2 \end{array}$$
(3)$$\begin{array}{c} n\;{\mathrm{CdI}}_{2}+n\;\mathrm{dpkoxH}\;\overset{\mathrm{MeCH}}{\to }\;\{[{\mathrm{CdI}}_{2}(\mathrm{dpkoxH})]{\}}_{n}\\ \;\;\;\;\;\;\;\;\;\;\;\;\;\;\;\;\;\;\;\;\;\;\;\;\;\;\;\;\;\;\;\;\;\;\;\;\;\;\;\;\;\;\;\;3 \end{array}$$

The yields were 50–60%. The three complexes were prepared using 1:2 Cd(II)/dpkoxH reaction ratios, i.e., in the presence of an excess of the ligand. Using the stoichiometric 1:1 ratio, the same complexes were again isolated (microanalytical and IR evidence) in comparable yields but with lower crystallinity. In an attempt to prepare 1:2 “CdX_2_(dpkoxH)_2_” compounds, we performed reactions with a larger excess of the ligand, i.e., 1:3, but again, compounds **1**∙2H_2_O, **2,** and **3** were isolated, suggesting that these complexes are thermodynamically stable under the reaction conditions employed.

### 2.2. Description of Structures

The structures of **1**∙2H_2_O, **2,** and **3** were determined via single-crystal X-ray crystallography. Crystallographic data are listed in Table 1. Selected interatomic distances and angles are given in Table 2. Various structural plots are shown in Figure 2, Figure 3, Figure 4 and Figure 5 and Appendix A. Further details can be found in the deposited CIF files. 

Compound **1**∙2H_2_O crystallizes in the centrosymmetric space group *P*ī. Complexes **2** and **3** are isomorphous, both crystallizing in the centrosymmetric space group *C*2/*c*; thus, only the crystal structure of **2** will be described in detail. The structures consist of 1D zigzag chains. In compound **1**∙2H_2_O, the chains extend along the *α* axis, while in complexes **2** and **3,** along the [1, 1, 0] direction. The zigzag motif is proven by the angle which is formed by three neighboring metal ions (Cd^II^∙∙∙Cd^II^∙∙∙Cd^II^), which is 115.7(1)° for **1**∙2H_2_O, 133.3(1)° for **2** and 130.8(1)° for **3**. Neighboring Cd^II^ atoms are alternately doubly bridged by halogeno and dpkoxH ligands which adopt the η^1^:η^1^:η^1^:μ (or 2.0111 using Harris notation [31]) coordination mode (Figure 6; see below). In each dimeric subunit, there is a crystallographically imposed inversion center. A terminal halogeno ligand completes 6-coordination at each metal ion. Thus, the coordination sphere of each Cd^II^ center is {Cd^II^(η^1^ − X)(μ − X)_2_(N_pyridyl_)_2_(N_oxime_)}.

The halogeno bridges are asymmetric, see Table 2; the {Cd(μ − X)_2_Cd} subunit is strictly planar due to symmetry. The Cd^II^∙∙∙Cd^II^ distances between the halogeno-bridged metal ions and the Cd^II^-X-Cd^II^ angles are, respectively, 3.859(1) Å and 94.1(1)° for **1**∙2H_2_O, 3.972(1) Å and 90.3(1)° for **2**, and 4.281(1) Å and 88.7(1)° for **3**. The Cd^II^∙∙∙Cd^II^ distance becomes larger as the size of the halogeno group increases (I^−^ > Br^−^ > Cl^−^). On the contrary, the distances between the dpkoxH-bridged Cd^II^ centers are practically the same (~6.0 Å) in the three complexes due to the similar type of bridging ligation. For a given complex, the Cd^II^-X_terminal_ bond length is shorter than the Cd^II^-X_bridging_ bond lengths, as expected. For example, the Cd1-Br1 bond length in **2** is 2.607(1) Å, while the Cd1-Br2 and Cd1-Br2* [1.5 − *x*, 1.5 − *y*, 1 − *z*] lengths are 2.643(1) and 2.950(1) Å, respectively (Figure 2b, Table 2). The two Cd^II^-N_pyridyl_ bond lengths in each compound differ significantly; the bond that is part of the 5-membered chelating ring is stronger than the bond to the nitrogen of the pyridyl ring that does not participate in chelation. For example, the Cd1-N1 and Cd1-N3** [1 − *x*, 1 − *y*, 1 − *z*] bond lengths in **3** are 2.375(6) and 2.503(7) Å, respectively (Appendix A, Table 2). The Cd^II^-N_pyridyl_ and Cd^II^-N_oxime_ bond lengths are typical for 6-coordinate metal cadmium(II) centers in 2-pyridyl oxime complexes [16,22,23,24,25,26,27].

The coordination geometry of the metal ion in the three complexes is distorted octahedral. The *trans*-donor–atom pairs are Cl_terminal_/N_oxime_ and two Cl_bridging_/N_pyridyl_ in **1**∙2H_2_O; on the contrary, the *trans* donor–atom pairs are X_terminal_/N_pyridyl_, X_bridging_/N_oxime_ and X_bridging_/N_pyridyl_ (X = Br, I) in the isomorphous complexes **2** and **3**. The *trans* angles are in the ranges 159.9(2)°–171.7(2)°, 155.2(1)–171.0(1)° and 155.6(2)–169.5(2)° in **1**∙2H_2_O, **2** and **3**, respectively. The distortion from the regular octahedral geometry arises primarily from the small bite angle of the 5-membered chelating “part” of the bridging dpkoxH ligand; the N1(pyridyl)-Cd^II^-N2(oxime) values are considerably less than the ideal value of 90°, i.e., 67.9(2)° for **1**∙2H_2_O and **2**, and 67.1(2)° for **3**.

There are two types of intrachain H-bonding interactions in the crystal structure of **1**∙2H_2_O. The first type is shown in Figure 2a, while both types are illustrated in Figure 3. In the first type, the donor is the carbon atom (C11) next to the pyridyl nitrogen atom N3 and the acceptor the terminal chloro group (Cl2); its dimensions are C11∙∙∙Cl2″ 3.54(1) Å, H11∙∙∙Cl2″ 2.803(2) Å and C11-H11∙∙∙Cl2″ 135.3(5)° [(″) 1 − *x*, 2 − *y*, −*z*]. In the second type, the donor is the oxime oxygen atom and the acceptor again the terminal chloro group; its dimensions are O1∙∙∙Cl2′ 3.106(5) Å, H(O1)∙∙∙Cl2′ 2.315(2) Å and O1-H(O1)∙∙∙Cl2′ 157.1(4)° [(′) 2 − *x*, 2 − *y*, −*z*]. The second type of intrachain H bond is also present in the structures of the isomorphous complexes **2** (Figure 2b and Appendix A) and **3** (Appendix A). The dimensions are O_oxime_∙∙∙Br_terminal_ 3.408(1) Å, H∙∙∙Br_terminal_ 2.699(1) Å, O_oxime_-H∙∙∙Br_terminal_ 143.1(1)° and O_oxime_-H∙∙∙I_terminal_ 3.581(5) Å, H∙∙∙I_terminal_ 2.945(1) Å, and O_oxime_-H∙∙∙I_terminal_ 136.6(4)°.

The packing of the chains in the structure of **1**∙2H_2_O is illustrated in Figure 4. The π-π stacking interactions between similar pyridyl rings of neighboring chains along the crystallographic axes *b* and *c* result in the formation of voids around the middle (0, 0.5, 0.5) of the base of the cell where the lattice H_2_O molecules are hosted. The 3D architecture of the structure is built through the π-π overlap of centrosymmetrically related pyridyl rings N1, C1, …, C5/N1, C1, …, C5 (1 − *x*, 3 − *y*, −*z*) and N3, C7, …, C11/N3, C7, …, C11 (1 − *x*, 2 − *y*, 1 − *z*) along *b* and *c*, respectively. The planes of the first pair of rings are at a distance of 3.41(2) Å, while those of the second one are at a distance of 3.38(2) Å.

The supramolecular characteristics of **2** and **3** are similar due to their isomorphism but different compared to those of **1**∙2H_2_O. Appendix A illustrates the interactions between neighboring chains for compound **2**. Neighboring chains are linked via pairs of C2-H2∙∙∙Cg1 interactions; C2 is an aromatic carbon of the N1-containing pyridyl ring, and Cg1 is the centroid of the N1, C1, …, C5 (0.5 + *x*, −0.5 + *y*, *z*) ring. Relevant dimensions are H2∙∙∙Cg1 2.731(1) Å, C2∙∙∙Cg1 3.592(2) Å and C2-H2∙∙∙Cg1 151.0(1) °. They also interact via pairs of π-π interactions between the N1, C1, …, C5 and N3, C7, …, C11 (1 − *x*, *y*, 0.5 − *z*) rings; the centroid–centroid distance is 4.392(1) Å and the angle that the planes form is 15.2(1)°. The former interactions among chains extending along the [1, 1, 0] direction result in the formation of layers parallel to the (001) plane (Appendix A), and the latter contributes to the 3D architecture of the structure. The layers are stacked along the *c* axis in a crossed-arrangement fashion relative to each other due to the 2-fold axis symmetry that relates the chains that belong to neighboring layers (Appendix A). The chains in the top layer shown in Appendix A are oriented along the [−1, 1, 0] direction. Layers of chains along the [1, 1, 0] or [−1, 1, 0] directions are stacked alternately along the *c* axis (Appendix A). As mentioned above, the 3D architecture of **3** is built via the same types of interactions. Geometric characteristics are H2∙∙∙Cg1 2.874(1) Å, C2∙∙∙Cg1 3.75(1) Å and C2-H2∙∙∙Cg1 154.1(6)°; in the pair of π-π interactions between the N1, C1, …, C5 and N3, C7, …, C11 (1 − *x*, *y*, 0.5 − *z*) rings, the centroid–centroid distance is 4.683(1) Å, and the angle between the planes is 15.8(2)°.

Concluding the structural part of this work, we already described the different supramolecular structures of **1**∙2H_2_O and **2**, **3**. However, careful inspection reveals that there is also a difference in the molecular conformation, despite the same formulae, the same coordination mode of dpkoxH, the similar coordination environment of the Cd^II^ centers, and the zigzag character of the chains; this difference is illustrated in the overlay diagram of Figure 5. The difference is also reflected (as reported earlier) in the Cd^II^∙∙∙Cd^II^∙∙∙Cd^II^ angles, which are smaller in **1**∙2H_2_O (115.7°) than in **2** and **3** (133.3° for **2** and 130.8° for **3**). We tentatively assign the differences in the molecular and supramolecular characteristics of the structures to the size of the halogeno ligands and the different nature of the pairs of the *trans* donor atoms between **1**∙2H_2_O and **2** and **3** (see above); the lattice H_2_O molecules in the chloro complex might also play a role, probably being engaged in H-bonding interactions.

Compounds **1**∙2H_2_O, **2,** and **3** are the first structurally characterized cadmium(II) complexes with the neutral or/and anionic forms of di-2-pyridyl ketone oxime as ligands. The observed 2.0111 (or η^1^:η^1^:η^1^:μ) coordination mode is very rare in the coordination mode of neutral dpkoxH and has been confirmed previously in three complexes (Table 3); the previously reported complexes [32,33,34] are dinuclear and, thus, the present compounds are the first characterized polymeric complexes with any metal that contain dpkoxH with 2.0111 ligations. Since many metal complexes of dpkoxH or/and dpkox^−^ are now known, we feel it is timely to collect the to-date crystallographically confirmed coordination modes of dpkoxH and dpkox^−^; these modes are illustrated in Figure 6.

We finalize this structural part by attempting to compare the present CdX_2_/dpkoxH complexes with their analogs of the other group 12 metal ions. The complexes [ZnCl_2_(dpkoxH)] (three polymorphs) [35,36] and [ZnBr_2_(dpkoxH)] [35] were characterized via single-crystal X-ray crystallography. The complexes consist of mononuclear (pseudo)tetrahedral molecules in which dpkoxH behaves as a N_pyridyl_, N’_pyridyl_-bidentate chelating ligand (1.0110 in Figure 6). This fact reflects the often-different coordination geometry between analogous Zn(II) and Cd(II) complexes. Comparison of **1**∙2H_2_O, **2,** and **3** with analogous Hg(II) complexes is not possible because HgX_2_/dpkoxH complexes are not known.

### 2.3. Spectroscopic Characterization in Brief

The complexes were characterized in the solid state using Raman and IR spectroscopies and in solution via NMR (^1^H, ^13^C, and ^113^Cd) techniques. To aid the solution studies, the molar conductivity values were also determined. Representative spectra are shown in Figure 7, Figure 8, Figure 9, Figure 10 and Figure 11 and Appendix A. It should be mentioned at the outset that most bands are not expected to be pure vibrations, and thus, the assignments represent approximate descriptions of the vibrational modes. 

The IR spectra of the complexes do not exhibit bands that are present in the free dpkoxH ligand, suggesting their purity. The presence of the neutral oxime group in **1**∙2H_2_O, **2,** and **3** by a broad band at ~3350 cm^−1^ assigned to *ν*(OH)_oxime_ [16]; the broadness of this band is indicative of hydrogen bonding, which has also been observed by crystallography. In the spectrum of **1**∙2H_2_O, the medium-intensity band at ~3430 cm^−1^ is attributed to the *ν*(OH) vibration of the lattice H_2_O [30]; its rather high wavenumber clearly reflects the non-coordinating nature of the water molecules. As expected, the O-H vibrations are hardly seen in the Raman spectra of the compounds. The strong IR band at 1636 cm^−1^ and the intense Raman peak at 1652 cm^−1^ in the spectra of free dpkoxH are assigned to the *ν*(C=N)_oxime_ vibration [36]. This band/peak in the spectra of the complexes is located at lower wavenumbers, suggesting oxime-N coordination [16]. The shift is rather small in the Raman spectra (the peaks appear at 1628–1644 cm^−1^) but large in the IR spectra where the strong band appears at 1595–1590 cm^−1,^ overlapping with an aromatic stretch of the 2-pyridyl groups [22]. The same trend is also evident in the *ν*(NO)_oxime_ mode, which appears at ~1015 (IR) and at approximately the same wavenumber (Raman) in the spectra of the complexes [37], lower than the wavenumber of this mode in the free ligand. The medium-intensity peaks in the Raman spectra at 3067 (**1**∙2H_2_O), 3066 (**2**), and 3060 (**3**) cm^−1^ are assigned to the stretching vibration of the aromatic carbon–hydrogen bonds, *ν*(CH) [38]; the corresponding IR band is weak and appears at 3060 cm^−1^ [36]. Exact assignments of the other 2-pyridyl modes are not an easy task because the spectra are complicated, and thus, any discussion about coordination shifts might be risky. The Raman peaks in the 415–210 cm^−1^ region would be associated with the Cd-X (X = Cl, Br, and I), Cd-N_oxime,_ and Cd-N_pyridyl_ stretching vibrations [38,39]. No X^−^-dependent peak was noticed, and the *ν*(CdX)_t_ and *ν*(CdX)_b_ modes (t = terminal; b = bridging) are most probably located below 210 cm^−1^, which is the low-frequency limit of the experiments.

In an attempt to probe the solution behavior of **1**∙2H_2_O, **2,** and **3**, the NMR spectra (^1^H, ^13^C, ^113^Cd) of the complexes in d_6_-DMSO were recorded at room temperature. The ^1^H and ^13^C NMR spectra of the three complexes are identical; however, the most remarkable feature is that the three spectra are also identical to the spectrum of free dpkoxH in the same solvent. This is evidence that the three complexes decompose in d_6_-DMSO, releasing dpkoxH, a fact attributable to the strong donor capacity of the solvent. The spectra show a singlet signal at δ 11.82 ppm assigned to the hydroxyl proton [16,30,36,40] and two doublets at δ 8.58 and 8.43 ppm attributed to the two non-equivalent H atoms of the aromatic carbons adjacent to the ring-N atoms [30,36]; the integration ratio of the three signals is 1:1:1, as expected. The six remaining aromatic protons appear as two multiple signals at δ 7.84 and 7.37 ppm and one doublet at δ 7.50 ppm with an integration ratio of 3:2:1, respectively.

The decomposition of the complexes in DMSO is corroborated by the ^13^C NMR spectra. The spectra of the three complexes and the free dpkoxH compound are identical. This means that the coordinated dpkoxH is released in solution, i.e., the solution Cd(II) species do not contain the oxime ligand. Thus, the ^13^C NMR spectra of the decomposition products are identical to those of the “free”, i.e., uncoordinated, dpkoxH compound. The spectra exhibit the 11 signals expected for the carbon atoms of the organic molecule in the δ 121.7–155.9 ppm region [36,41,42,43,44]. The signal at δ 155.2 ppm is due to the carbon atom of the oxime group [36]; exact assignments of the pyridyl ^13^C resonances would be risky.

In an attempt to probe the solution Cd(II) species in more detail, we recorded the ^113^Cd NMR spectra of the complexes in d_6_-DMSO, combined with molar conductivity values (25 °C, 10^−3^ M) in DMSO. ^113^Cd NMR spectroscopy is a diagnostic tool for Cd(II) species in solution. The reasons are the nuclear spin I of ½ and its relatively good receptivity (R = 7.59 relative to ^13^C (R = 1)) [45]. It should be mentioned at this point that there is a danger of oversimplification of the chemical shift data. This is because chemical exchange often occurs between different complexation or binding sites (including ligand and solvent) for Cd(II), and the observed line is a time average of the different chemical shifts from these different sites [45,46]. The Λ_Μ_ values of **1**∙2H_2_O and **2** are 71 and 66 S cm^2^ mol^−1^, respectively, suggesting 1:2 electrolytes [47]. This is strong evidence that chlorides and bromides do not participate in coordination with Cd(II) in solution. On the contrary, the Λ_Μ_ value of **3** is 5 S cm^2^ mol^−1,^ indicating a negligible ionization in solution, i.e., a non-electrolyte behavior [47]. This fact suggests that the iodides are bonded to Cd(II) in solution. In analogy, the ^13^Cd NMR spectra of **1**∙2H_2_O and **2** in d_6_-DMSO are identical and different from the corresponding spectrum of **3**. The spectra of the two former complexes consist of a single resonance at δ 196.7 ppm, while the spectrum of the iodo complex displays again one signal at a completely different δ value (77.2 ppm). The appearance of one signal indicates that there is one Cd(II) species in solution for every complex. Moreover, this spectral behavior shows that the species of **1**∙2H_2_O and **2** are identical and different from the species arising from **3**. The δ values indicate Cd(II) species in a predominantly O-environment [48]. The δ values of 77.2 ppm for complex **3** indicate the coordination of iodo ligands to Cd^II^ [45] since values of +43 and +101 ppm have been recorded in Cd(II) iodo complexes in solution. Taken together, the above molar conductivity and spectral data suggest that the decomposition of the complex most probably occurs via the processes represented by Equations (4)–(6), where DMSO is O-bonded.
(4)$$\begin{array}{c} \{[{\mathrm{CdCl}}_{2}(\mathrm{dpkoxH})]\cdot 2H_{2}O{\}}_{n}+6n\;\mathrm{DMSO}\;\overset{\mathrm{DMSO}}{\to }n\;[Cd(DMSO{)}_{6}{]}^{2+}+2n\;Cl^{-}+n\;dpkoxH+2n\;H_{2}O\\ \;\;\;\;\;\;\;\;\;\;\;\;\;\;\;\;\;\;\;\;\;\;\;\;1\cdot 2H_{2}O \end{array}$$
(5)$$\begin{array}{c} \{[{\mathrm{CdBr}}_{2}(\mathrm{dpkoxH})]{\}}_{n}+6n\;\mathrm{DMSO}\;\overset{\mathrm{DMSO}}{\to }\;n\;[Cd(DMSO{)}_{6}{]}^{2+}+2n\;Br^{-}+n\;dpkoxH\\ \;\;\;\;\;\;\;\;\;\;\;\;\;\;\;\;\;\;\;\;2 \end{array}$$
(6)$$\begin{array}{c} \{[{\mathrm{CdI}}_{2}(\mathrm{dpkoxH})]{\}}_{n}+4n\;\mathrm{DMSO}\;\overset{\mathrm{DMSO}}{\to }n\;[CdI_{2}(DMSO{)}_{4}]+n\;dpkoxH\\ \;\;\;\;\;\;\;\;\;\;\;\;\;\;\;\;\;\;\;3 \end{array}$$

The different solution behavior can be attributed to the nature of the halogeno ligand. The Cd^II^-Cl and Cd^II^-Br coordination bonds have a higher polar contribution than the Cd^II^-I bonds, which should be more covalent. Thus, the chloro and bromo 1D polymers release the halogeno ligand from the coordination sphere of Cd^II^ easier than **3**, which keeps the iodo groups bonded with the metal ion more tightly. The different coordination behavior of Cl^−^, Br^−,^ and I^−^ ligands in Cd(II)/DMSO complexes has also been observed and discussed in solid-state structures [49].

Efforts to further study the solution behavior of the complexes via ESI-MS were unsuccessful. The compounds are insoluble in the common ESI-MS solvents (MeCN, MeOH, H_2_O) but also in DMSO:MeOH (1:9 *v*/*v*) mixtures.

## 3. Experimental Section

### 3.1. Materials and Instrumentation

All manipulations were performed under aerobic conditions. Deionized water was received from the in-house facility. Solvents and reagents were purchased from Sigma-Aldrich (Tanfrichen, Germany) and Alfa Aesar (Karlsruhe, Germany) and used as received. The purity of dpkoxH was checked by ^1^H NMR spectroscopy. **Safety note:** Cd(II) compounds are toxic, and small quantities should be used; the use of gloves is recommended.

Microanalyses (C, H, and N) were performed by the Instrumental Analysis Center of the University of Patras. Conductivity measurements in DMSO were carried out at room temperature (24–26 °C) with a Metrohm-Herisau E-527 bridge and a cell of standard design; the concentration of the solutions was ~10^−3^ M. FT-IR spectra (4000–400 cm^−1^) were recorded using a Perkin-Elmer 16PC spectrometer (Waltham, MA, USA); the samples were in the form of KBr pellets. For the Raman measurements, the T64000 Horiba Jobin Yvon micro-Raman setup was used. The excitation wavelength was 514.5 nm emitted from a DPSS laser (Cobolt Fandango TMISO laser, Norfolk, UK). The laser power on the sample was 2.5 mW. The backscattered radiation was collected from a single configuration of the monochromator after passing through an appropriate edge filter (LP02-633RU-25, laser2000 UK Ltd., Huntingdon, Cambridgeshire, UK). The calibration of the instrument was achieved via the standard Raman peak position of Si at 520.5 cm^−1^. The spectral resolution was 5 cm^−1^. ^1^H and ^13^C NMR spectra in d_6_-DMSO were recorded on a Bruker Avance DPX spectrometer (Bruker AVANCE, Billerica, MA, USA) at resonance frequencies of 400.13 MHz (^1^H) and 100.62 MHz (^13^C); Me_4_Si was used as an internal standard. ^113^Cd NMR spectra in d_6_-DMSO were recorded on a Bruker Ascend (600 MHz) spectrometer; the reference was an aqueous Cd(ClO_4_)_2_∙6H_2_O 0.1 M solution (δ = 0 ppm).

### 3.2. Preparation of the Complexes

*{[CdCl_2_(dpkoxH)]∙2H_2_O}_n_ (***1***∙2H_2_O):* A solution of CdCl_2_∙2H_2_O (0.022 g, 0.10 mmol) in MeOH (8 mL) was added to a solution of dpkoxH (0.039 g, 0.20 mmol) in the same solvent (2 mL). The resulting colorless solution was stirred for 10 min and stored in a closed vial at 25 °C. X-ray quality, colorless crystals of the product were precipitated within 48 h. The needle-like crystals were collected via filtration, washed with ice-cold EtOH (1 mL), and Et_2_O (2 × 1 mL), and dried in air. The yield was 54% (based on the metal ion available). Anal. Calcd. (%) for C_11_H_13_CdCl_2_N_3_O_3_: C, 31.56; H, 3.14; N, 10.04. Found (%): C, 31.41; H, 3.63; N, 9.89. IR (KBr, cm^−1^): 3430m, 3315sb, 3062w, 1620w, 1594m, 1560m, 1534w, 1474m, 1436m, 1408w, 1374m, 1332m, 1292w, 1202w, 1166w, 1064s, 1044s, 1018s, 1006sh, 976w, 788m, 752m, 688w, 676w, 628m, 598w, 570w, 474w, 406w. Raman (cm^−1^): 3067m, 1628s, 1597m, 1569s, 1483s, 1445w, 1371s, 1337w, 1299w, 1206m, 1056m, 1009s, 790w, 745w, 687w, 628w, 414w, 325w, 233m. ^1^H NMR (d_6_-DMSO, δ/ppm): 11.81 (s, 1H), 8.61 (d, 1H, *J* = 2.9 Hz), 8.47 (d, 1H, *J* = 2.9 Hz), 7.82 (mt, 3H), 7.53 (d, 1H, *J =* 5.2 Hz), 7.40 (mt, 2H). ^13^C NMR (d_6_-DMSO, δ/ppm): 122.3, 123.9, 124.2, 126.0, 136.7, 137.7, 149.3, 149.6, 152.5, 154.7, 155.2. ^113^Cd NMR (d_6_-DMSO, δ/ppm): 196.8. Λ_Μ_ (DMSO, 10^−3^ M, 25 °C) = 71 S cm^2^ mol^−1^.

*{[CdBr_2_(dpkoxH)]}_n_ (***2***):* A solution of CdBr_2_∙4H_2_O (0.034 g, 0.10 mmol) in MeCN (25 mL) was added to a solution of dpkoxH (0.040 g, 0.20 mmol) in the same solvent (10 mL). The resulting colorless solution was stirred for 20 min and allowed to slowly evaporate at room temperature. X-ray quality, colorless crystals of the product were obtained within 3 d. The plate-like crystals were collected by filtration, washed with ice-cold EtOH (2 × 1 mL) and Et_2_O (4 × 2 mL), and dried in air. The yield was 53% (based on the cadmium available). Anal. Calcd. (%) for C_11_H_9_CdBr_2_N_3_O_3_: C, 28.02; H, 1.93; N, 8.92. Found (%): C, 27.90; H, 2.15; N, 9.14. IR (KBr, cm^−1^): 3342mb, 3280sh, 3058w, 1618w, 1590s, 1566m, 1474m, 1438m, 1408w, 1362w, 1330s, 1292w, 1198m, 1158w, 1094w, 1062s, 1042s, 1016s, 1006sh, 966m, 794m, 752m, 742m, 686m, 672m, 628m, 594w, 540wb, 466w, 408m. Raman (cm^−1^): 3097m, 3066s, 3027w, 1644s, 1593s, 1571s, 1476s, 1441m, 1410m, 1333m, 1303w, 1290w, 1268w, 1202s, 1159w, 1110w, 1058m, 1017s, 1006sh, 799w, 785w, 748w, 689w, 639w, 589w, 407w, 329w, 238w, 219m. ^1^H NMR (d_6_-DMSO, δ/ppm): 11.81 (s, 1H), 8.58 (d, 1H, *J* = 2.8 Hz), 8.43 (d, 1H, *J* = 2.8 Hz), 7.84 (mt, 3H), 7.50 (d, 1H, *J* = 5.2 Hz), 7.37 (mt, 2H). ^13^C NMR (d_6_-DMSO, δ/ppm): 122.4, 123.8, 124.0, 125.9, 136.8, 137.8, 149.2, 149.5, 152.5, 154.8, 155.3. ^113^Cd NMR (d_6_-DMSO, δ/ppm): 196.3. Λ_Μ_ (DMSO, 10^−3^ M, 25 °C) = 66 S cm^2^ mol^−1^.

*{[CdI_2_(dpkoxH)]}_n_ (***3***):* A solution of CdI_2_ (0.036 g, 0.10 mmol) in MeOH (3 mL) was added to a solution of dpkoxH (0.039 g, 0.20 mmol) in the same solvent (7 mL). The resulting colorless solution was stored in a closed vial for 30 d. During this time, the color of the solution slowly became pale yellow, and after one month (total of 60 d), X-ray quality, colorless plate-like crystals of the product were precipitated. The crystals were collected via filtration, washed with ice-cold EtOH (2 × 1 mL) and Et_2_O (3 × 2 mL), and dried in air. The yield was 51% (based on the cadmium available). Anal. Calcd. (%) for C_11_H_9_CdI_2_N_3_O: C, 23.37; H, 1.61; N, 7.43. Found (%): C, 23.87; H, 1.69; N, 7.09. IR (KBr, cm^−1^): 3355mb, 3065w, 1614w, 1590m, 1560m, 1474m, 1438s, 1398m, 1330w, 1286w, 1198w, 1062sh, 1040s, 1014m, 960w, 899w, 880w, 796m, 776m, 758m, 690m, 668s, 636sh, 626m, 586w, 550m, 496m, 470w, 408m. Raman (cm^−1^): 3087w, 3060s, 1642s, 1593s, 1571s, 1476s, 1441m, 1402s, 1333m, 1304w, 1290w, 1202s, 1058m, 1015s, 781w, 750w, 693w, 592w, 407w, 330w, 238w, 219m. ^1^H NMR (d_6_-DMSO, δ/ppm): 11.82 (s, 1H), 8.58 (d, 1H, *J* = 2.8 Hz), 8.43 (d, 1H, *J* = 2.8 Hz), 7.83 (mt, 3H), 7.50 (d, 1H, *J* = 5.2 Hz), 7.37 (mt, 2H). ^13^C NMR (d_6_-DMSO, δ/ppm): 121.9, 123.7, 124.0, 125.8, 136.4, 137.3, 149.2, 149.5, 152.4, 154.9, 155.2. ^113^Cd NMR (d_6_-DMSO, δ/ppm): 77.2. Λ_Μ_ (DMSO, 10^−3^ M, 25 °C) = 5 S cm^2^ mol^−1^.

### 3.3. Single-Crystal X-ray Crystallography

Colorless crystals of **1**∙2H_2_O and **3** were taken directly from the mother liquor and immediately cooled to −103(2) and −93(2) °C, respectively. Diffraction data for these compounds were collected on a Rigaku R-AXIS Image Plate diffractometer (Rigaku Americas Corporation, The Woodlands, TX, USA) using graphite-monochromated Cu Κα radiation. Data collection (ω-scans) and processing (cell refinement, data reduction, and empirical/numerical absorption correction) were performed using the CrystalClear program package [50]. The structures were solved by direct methods using SHELXS, ver. 2013/1 [51] and refined by full-matrix least-squares techniques on *F*^2^ with SHELXL, ver. 2014/6 [52]. The H atoms were located by difference maps and refined isotropically. All non-H atoms were refined anisotropically. For the chloro compound, the SQUEEZE procedure [53] was applied, and an estimated number of two lattice H_2_O molecules per formula unit was derived based on the accessible solvent voids volume or electron counts within the voids volume. The complex has been abbreviated as **1**∙2H_2_O throughout this work. Diffraction data for complex **2** were collected with a Bruker APEX II Quasar diffractometer (Bruker Analytical X-Ray Systems, Madison, WI, USA), equipped with a graphite monochromator on the path of Mo Kα radiation. Crystals, taken directly from the reaction solution, were coated with Cargille^TM^ NHV immersion oil and mounted on a fiber loop, followed by data collection at −153 °C. The program SAINT was used to integrate the data, which were thereafter corrected using SADABS [54]. The structure was solved using SHELXT [55] and refined by the full-matrix least-squares technique on *F*^2^ using SHELXL-2018 [52]. All H atoms were located by difference maps and refined isotropically; all non-H atoms were refined with anisotropic displacement parameters. Plots of the structures were drawn using the Diamond 3 program package [56].

Crystallographic data were deposited with the Cambridge Crystallographic Data Centers, Nos 2311945 (**1**∙2H_2_O), 2311946 (**2**), and 2311947 (**3**). Copies of the data can be obtained free of charge upon application to CCDC, 12 Union Road, Cambridge, CB2 1EZ, UK: Tel.: +(44)-1223-762910; Fax: +(44)-1223-336033; E-mail: deposit@ccdc.cam.ac.uk.

## 4. Conclusions in Brief and Perspectives

In this work, we reported the employment of dpkoxH in reactions with Cd(II) halides which has resulted in the isolation of three 1D polymeric complexes. Thus, the first Cd(II)/dpkoxH complexes have been prepared and characterized. The most important features of this work are: (a) The reactions lead to 1D zigzag polymers with a 1:1 metal-to-ligand ratio; no monomeric 1:2 complexes could be obtained. (b) The three complexes have interesting molecular and supramolecular structures, with the neutral ligand exhibiting a rare coordination mode. (c) The nature of the halogeno ligand slightly affects the molecular structure and significantly the supramolecular characteristic, with the bromo and iodo complexes being isomorphous. Isostructurality among the non-fluorine halogens is a well-documented phenomenon in organic chemistry [57]. However, there are some limits to this kind of halogen equivalence in supramolecular organic chemistry. In molecular and supramolecular inorganic chemistry, however, non-equivalence is common [58]. This has been attributed to size, electronegativity, and polarization effects. Contrary to the general trend according to which the chloro and bromo ligands play the same role in the crystal structures, whereas the iodo ligand is distinct [58], we have observed a different trend in the present work; complexes **2** and **3** are isomorphous, whereas **1**∙2H_2_O is different. The increase in the size of the halogeno ligand results in the distortion of the Cd^II^ coordination sphere by increasing the X_bridging_-Cd^II^-X_bridging_ angle (85.9° for **1**∙2H_2_O, 86.7° for **2**, 91.3° for **3**); at the same time, the two X_bridging_-Cd^II^-X_terminal_ angles are significantly smaller in the chloro complex (103.2° and 87.3° in **1**∙2H_2_O, compared to 112.1°, 95.4° for **2**, and 110.0°, and 93.2° for **3**). These variations, in turn, affect other coordination bond angles and eventually the nature of the *trans*-donor pairs, which are similar in **2** and **3** and different from that in **1**∙2H_2_O (see structural descriptions in Section 2.2). The differences in the molecular characteristics affect the supramolecular features; the presence of lattice H_2_O in the chloro complex might also play a role. (d) The complexes do not retain their solid-state structures in DMSO solution, and again the nature of the halogeno ligand affects the species identity; contrary to the solid state, the chloro and bromo complexes have an identical behavior in solution compared to that of the iodo compound; and (e) Comparison of the coordination chemistry of neutral dpkoxH toward CdX_2_ and ZnX_2_ (X = Cl, Br, …) reveals differences. Although the stoichiometry in both cases is 1:1, the Zn(II) complexes are monomeric tetrahedral [35,36], whereas the Cd(II) ones are polymeric octahedral, reflecting the well-known differences in the chemistry of these group 12 metal ions.

Comparison of the present Cd(II)/dpkoxH complexes with Cd(II) complexes of other 2-pyridyl *ald*oximes and *ket*oximes is not direct; the reason is that dpkoxH has another site capable of coordination. In all the previously characterized complexes, the neutral 2-pyridyl oxime molecule behaves as a N_pyridyl_ and N_oxime_ -bidentate chelating ligand. The mononuclear [16,26,27,29,30] and dinuclear [25] complexes have a 1:2 or 1:3 metal-to-ligand stoichiometry. As mentioned above, mononuclear or dinuclear Cd(II)/dpkoxH halide complexes could not be isolated. Most of the polymeric Cd(II)/2-pyridyl oxime complexes have a 1:1 metal-to-ligand stoichiometry [22,26,28,30], with notable exceptions being the 1D coordination polymers {[Cd(fum)(2-pyaoH)_2_]}_n_ [22] and {[Cd(suc)(2-pyaoH)_2_]}_n_ [25]. In all polymeric Cd(II)/2-pyridyl oxime compounds, bridging of the neighboring Cd^II^ atoms is achieved via the ancillary dicarboxylate [22,25,26], sulfate [26,28] or halogeno [30] ligands. In the Cd(II)/dpkoxH complexes of the present work (**1**∙2H_2_O, **2**, **3**), the linking of neighboring Cd^II^ atoms is attained via alternate bis(halogeno) and bis(dpkoxH) bridges; this behavior is the result of the second 2-pyridyl nitrogen which allows the 2.0111 coordination mode (Figure 6).

A possible scenario for the formation of the polymeric Cd(II) complexes is the following: First, a monomeric “CdX_2_(dpkoxH)” complex (with terminal halogeno ligands, and a N_pyridyl_, N_oxime_-bidentate chelating dpkoxH molecule) or a dimeric “Cd_2_X_4_(dpkoxH)_2_” species (either with exclusively terminal halogeno groups and two N, N′, and N″-bridging dpkoxH molecules, or with both terminal and bridging halogeno groups and two N_pyridyl_ and N_oxime_-bidentate chelating dpkoxH molecules) are formed in solution. The Cd^II^ atoms are coordinatively unsaturated in these species (coordination numbers 4 or 5) and then polymerization occurs during the precipitation of the products. 

With the knowledge and experience obtained in this work, our future research efforts are directed, among others, to (1) the enrichment of the Cd(II) position in the “Periodic Table” of dpkoxH by using other Cd(II) sources, such as nitrate, perchlorate, sulfate, and carboxylates, and by adding an external base (e.g., Et_3_N, LiOH, and Bu_4_^n^NOH) in the reaction systems in order to obtain complexes (clusters and coordination polymers) containing the deprotonated dpkox^−^ ligand; the expected coordination of the deprotonated oximate oxygen atom gives enormous possibilities for different and interesting chemistry. (2) The preparation of HgX_2_/dpkoxH (X = Cl, Br, I) complexes with the goal of discovering similarities or/and differences with the corresponding Cd(II) compounds; and (3) The study of the reactions between CdX_2_ (X = Br, I) and phenyl 2-pyridyl ketoxime (phpaoH; R = Ph in Figure 1) and comparison of the products with the complexes of the present work; the ligands dpkoxH and phpaoH have identical steric properties and rather similar electronic properties, but they differ in the number of donor atoms (phpaoH lacks the second 2-pyridyl N atom). Our previous work [30] on the CdCl_2_/phpaoH reaction system revealed the existence of both monomeric [CdCl_2_(phpaoH)_2_] and polymeric {[CdCl_2_(phpaoH)]}_n_ (with a different structure, as expected, compared with the structures of **1**∙2H_2_O) compounds and this also gives hopes for interesting CdX_2_/phpaoH (X = Br, I) chemistry. Some of our efforts along the three above-mentioned topics are already well advanced, and the results will be submitted soon.

## Figures and Tables

**Figure 1 molecules-29-00509-f001:**
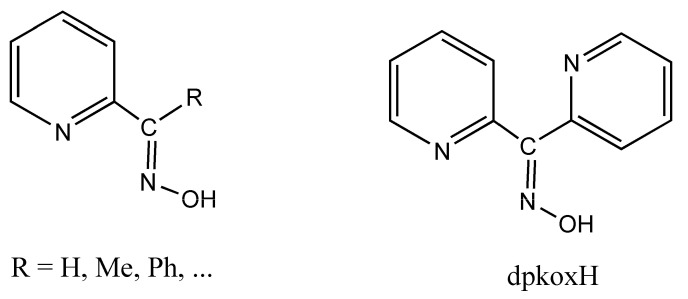
(**Left**) The general structural formula of simple 2-pyridyl oximes (R is a non-donor group); (**right**) di-2-pyridyl ketone oxime (dpkoxH), the ligand used in this work.

**Figure 2 molecules-29-00509-f002:**
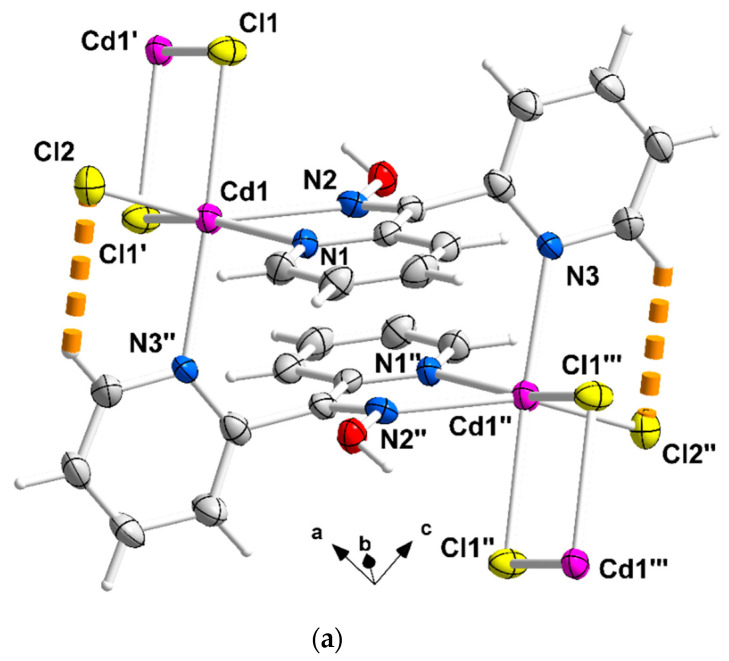
Complete coordination spheres of two neighboring Cd^II^ centers in the zigzag chains of complex **1**∙2H_2_O (**a**) and **2** (**b**). Symmetry codes: (′) 2 − *x*, 2 − *y*, −*z*; (″) 1 − *x*, 2 − *y*, −*z*; (‴) −1 + *x*, *y*, *z* for complex **1**∙2H_2_O, and (*) 1.5 − *x*, 1.5 − *y*, 1 − *z*; (**) 1 − *x*, 1 − *y*, 1 − *z*); (***) −0.5 + *x*, −0.5 + *y*, *z* for compound **2**. The thick dashed orange lines represent C-H∙∙∙Cl_terminal_ (**a**) and O_oxime_-H∙∙∙Br_terminal_ (**b**) H bonds.

**Figure 3 molecules-29-00509-f003:**
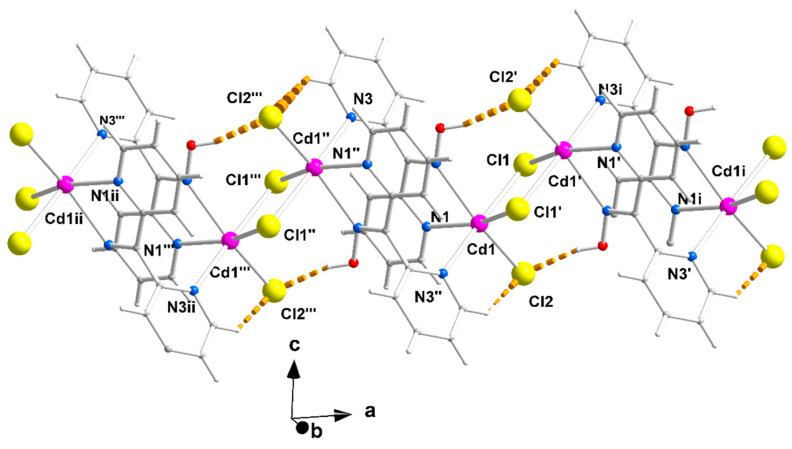
A portion of one zigzag chain (developed along the *α* axis) of complex **1**∙2H_2_O; the lattice H_2_O molecules are not drawn. The thick dashed orange lines represent intrachain C-H∙∙∙Cl_terminal_ and O_oxime_-H∙∙∙Cl_terminal_ H bonds. The symmetry codes (′), (″), and (‴) are the same as those described in Figure 2a, and the additional codes are (i) 1 + *x*, *y*, *z* and (ii) −*x*, 2 − *y*, *z*.

**Figure 4 molecules-29-00509-f004:**
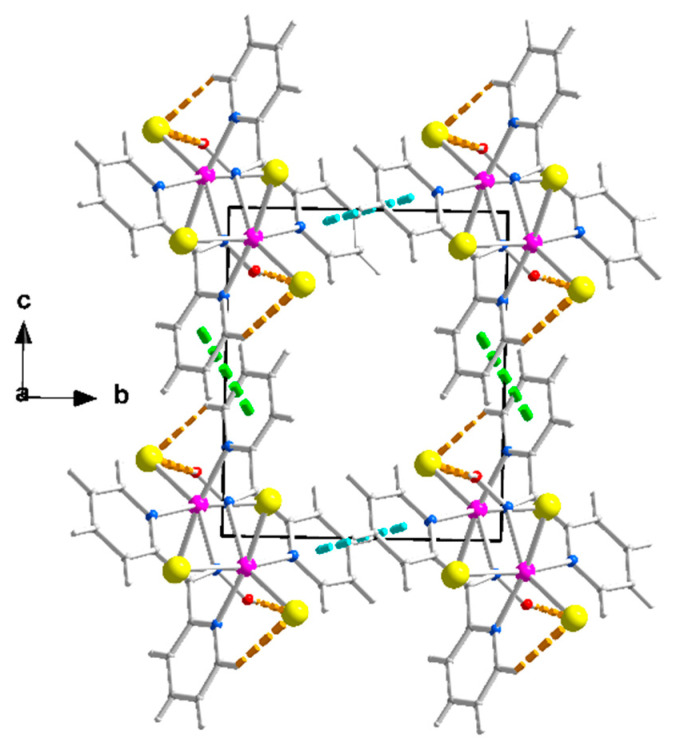
Packing of chains in the crystal structure of complex **1**∙2H_2_O. The π-π stacking interactions between the rings N1, C1, …, C5 and N1, C1, …, C5 (1 − *x*, 3 − *y*, −*z*), and between the rings N3, C7, …, C11 and N3, C7, …, C11 (1 − *x*, 2 − *y*, 1 − *z*) are indicated with thick cyan and light green lines, respectively. The thick dashed orange lines represent the intrachain H bonds; see Figure 3.

**Figure 5 molecules-29-00509-f005:**
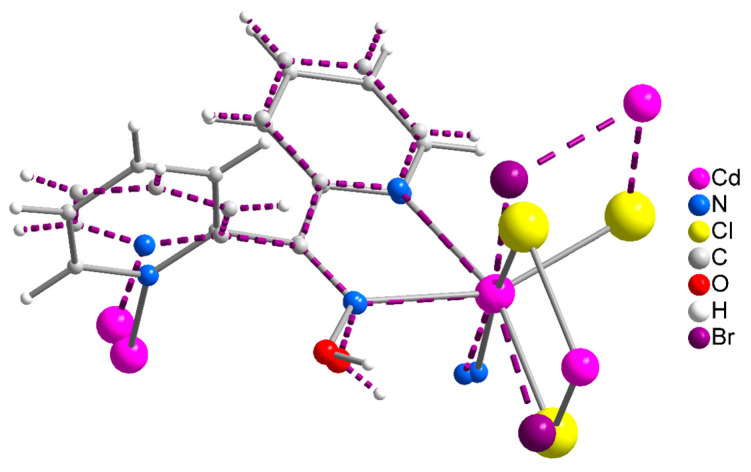
A very small portion of the chains of complexes **1** and **2** in the overlay mode. The solid lines correspond to **1,** and the dashed lines to **2**. The color scheme of the atoms involved is shown on the right.

**Figure 6 molecules-29-00509-f006:**
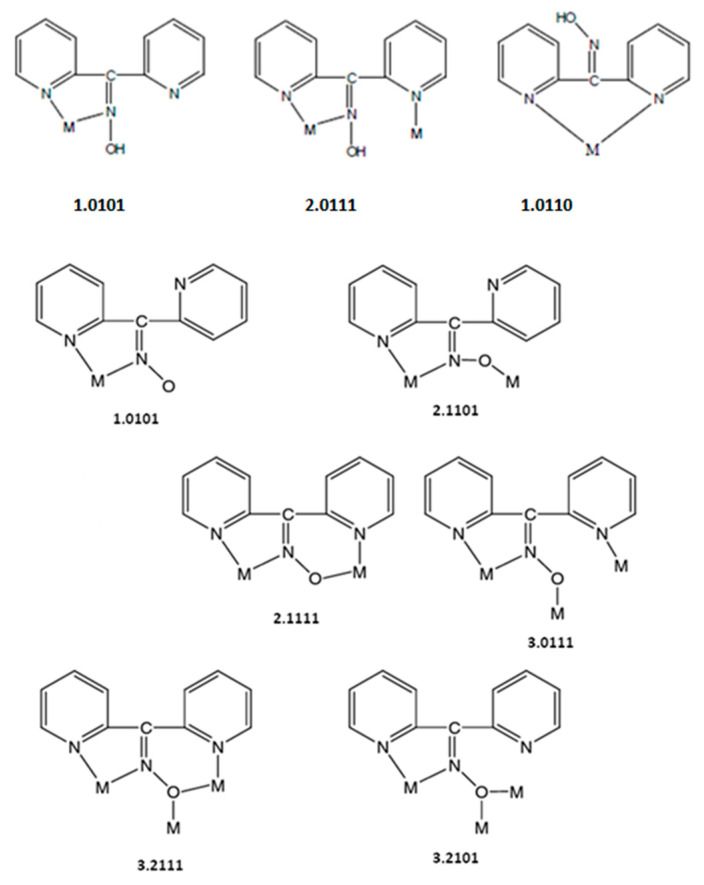
The to-date crystallographically confirmed coordination modes of dpkoxH and dpkox^−^, and the Harris notation that describes these modes; the ligation mode observed in **1**∙2H_2_O, **2,** and **3** is shown in the middle of the upper row.

**Figure 7 molecules-29-00509-f007:**
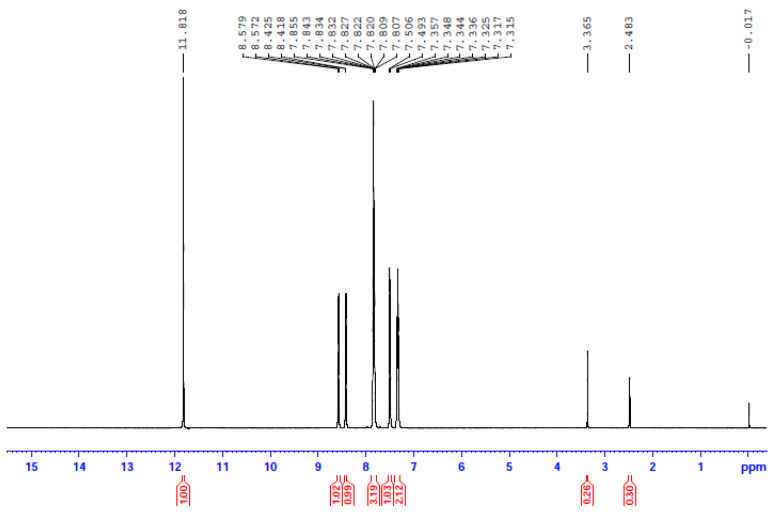
The ^1^H NMR spectrum of **3** in d_6_-DMSO. The signal at δ 2.48 ppm is due to the methyl groups of the non-deuterated amount of the solvent, and the signal at δ 3.37 ppm to the protons of the H_2_O content of the solvent.

**Figure 8 molecules-29-00509-f008:**
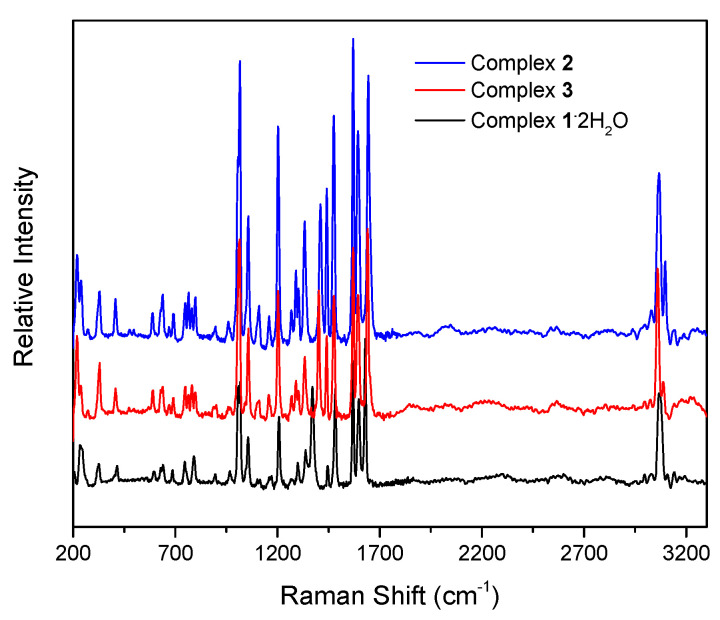
The Raman spectra of complexes **1**∙2H_2_O (black), **2** (blue), and **3** (red).

**Figure 9 molecules-29-00509-f009:**
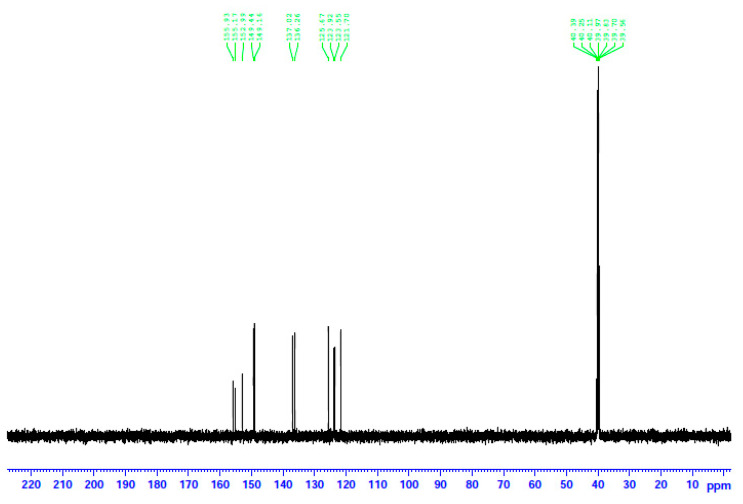
The ^13^C NMR spectrum of **2** in d_6_-DMSO. The signals at δ ~40 ppm are due to the methyl carbon atoms of the solvent.

**Figure 10 molecules-29-00509-f010:**
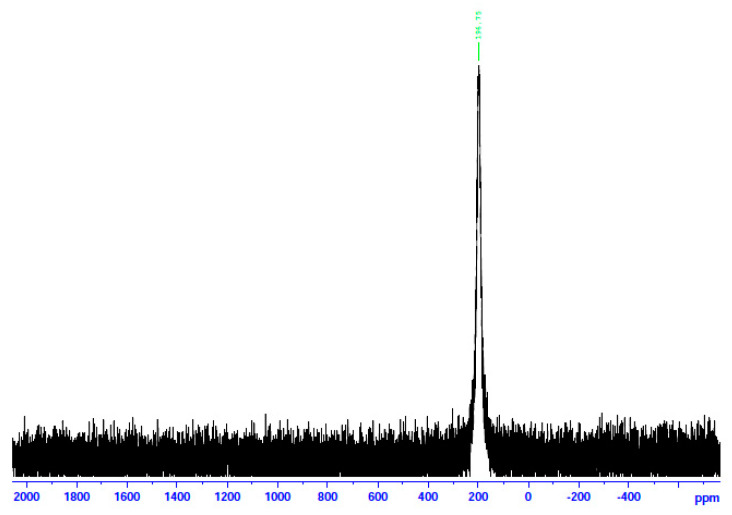
The ^113^Cd NMR spectrum of **1**∙2H_2_O in d_6_-DMSO.

**Figure 11 molecules-29-00509-f011:**
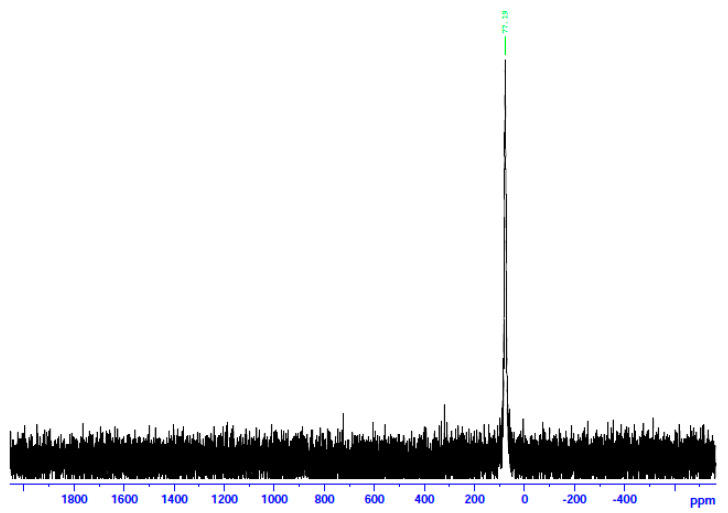
The ^113^Cd NMR spectrum of **3** in d_6_-DMSO.

**Table 1 molecules-29-00509-t001:** Crystallographic data and refinement parameters for compounds **1**∙2H_2_O, **2,** and **3**.

Parameter	{[CdCl_2_(dpkoxH)]∙2H_2_O}_n_ (1∙2H_2_O)	{[CdBr_2_(dpkoxH)]}_n_ (2)	{[CdI_2_(dpkoxH)]}_n_ (3)
Empirical formula	C_11_H_9_CdCl_2_N_3_O∙2H_2_O	C_11_H_9_CdBr_2_N_3_O	C_11_H_9_CdI_2_N_3_O
Formula weight	418.54	471.43	565.41
Crystal system	Triclinic	Monoclinic	Monoclinic
Space group	*P*ī	*C*2/*c*	*C*2/*c*
Color	Colorless	Colorless	Colorless
*a*, Å	8.3231(2)	15.8842(11)	16.0480(12)
*b*, Å	9.0416(2)	9.2829(6)	9.6812(6)
*c*, Å	10.4143(2)	18.0648(12)	18.7611(13)
*α*, °	88.371(1)	90.00	90.00
*β*, °	83.666(1)	93.572(2)	93.558(5)
*γ*, °	76.119(1)	90.00	90.00
Volume, Å^3^	756.19(3)	2658.5(3)	2909.2(3)
*Z*	2	8	8
Temperature, °C	−103	−153	−93
Radiation, Å	Cu *K*α, 1.54178	Μο *Κ*α, 0.71073	Cu *Κ*α, 1.54178
Calculated density, g∙cm^−3^	1.838	2.356	2.582
Absorption coefficient, mm^−1^	14.92	7.64	45.30
No. of measured, independent, and observed [*I* > 2*σ*(*I*)] reflections	17,055, 2607, 2400	49,909, 2736, 2684	12,775, 2518, 2269
*R* _int_	0.092	0.030	0.113
Final *R* indices [*I* > 2*σ*(*I*)] ^α^	*R*_1_ = 0.0641 *wR*_2_ = 0.1625	*R*_1_ = 0.0129 *wR*_2_ = 0.0325	*R*_1_ = 0.0605 *wR*_2_ = 0.1601
Number of parameters	164	164	164
Goodness-of-fit on *F*^2^	1.15	1.07	1.06
Larger differences in peak and hole (e Å^−3^)	1.95/−1.67	0.38/−0.46	1.30/−1.91

^α^ *R*_1_ = Σ(|*F*_o_| − |*F*_c_|)/Σ(|*F*_o_|), *wR*_2_ = {Σ[*w*(*F*_o_^2^ − *F*_c_^2^)^2^]/Σ[*w*(*F*_o_^2^)^2^]}^1/2^, *w* = 1[*σ*^2^(*F*_o_^2^) + (α*P*)^2^ + b*P*], where *P* = [max(*F*_o_^2^, 0) + 2*F*_c_^2^]/3 (a = 0.1041 and b = 1.5496 for **1**∙2H_2_O; a = 0.0163 and b = 4.1924 for **2**; a = 0.0984 and b = 0.2896 for **3**).

**Table 2 molecules-29-00509-t002:** Selected bond lengths (Å) and angles (°) for the polymeric complexes **1**∙2H_2_O, **2,** and **3**.

Complex 1∙2H_2_O ^a^		Complex 2 ^b^		Complex 3 ^c^	
		**Lengths (** **Å)**			
Cd1-Cl1	2.731(2)	Cd1-Br2*	2.950(1)	Cd1-I2*	3.290(1)
Cd1-Cl2	2.520(2)	Cd1-Br2	2.643(1)	Cd1-I2	2.818(1)
Cd1-Cl1′	2.539(2)	Cd1-Br1	2.607(1)	Cd1-I1	2.787(1)
Cd1-N1	2.314(5)	Cd1-N2	2.402(1)	Cd1-N2	2.437(7)
Cd1-N2	2.473(6)	Cd1-N1	2.355(1)	Cd-N1	2.375(6)
Cd1-N3″	2.450(5)	Cd1-N3**	2.486(1)	Cd1-N3**	2.503(7)
		**Angles (°)**			
Cl1-Cd1-Cl2	87.3(1)	Br2*-Cd1-Br2	86.7(1)	I2*-Cd1-I2	91.3(2)
Cl1-Cd1-Cl1′	85.9(1)	Br2*-Cd1-Br1	95.4(1)	I2*-Cd1-I1	93.2(1)
Cl1-Cd1-N2	76.3(1)	Br2*-Cd1-N2	75.8(1)	I2-Cd1-N2	71.6(2)
Cl1-Cd1-N1	101.0(1)	Br2*-Cd1-N1	83.1(1)	I2*-Cd1-N1	81.6(2)
Cl2-Cd1-Cl1′	103.2(1)	Br2-Cd1-Br1	112.1(1)	I2-Cd1-I1	110.0(1)
Cl1′-Cd1-N2	95.9(2)	Br1-Cd1-N2	87.7(1)	I1-Cd1-N2	88.8(2)
N2-Cd1-N1	67.9(2)	N2-Cd1-N1	67.9(1)	N2-Cd1-N1	67.1(2)
N1-Cd1-Cl2	96.1(2)	N1-Cd1-Br2	92.7(1)	N1-Cd1-I2	93.8(1)
N3″-Cd1-Cl2	95.3(2)	N3**-Cd1-Br2	90.7(1)	N3**-Cd1-I2	92.4(1)
N3″-Cd1-Cl1′	85.8(2)	N3**-Cd1-Br1	92.8(1)	N3**-Cd1-I1	94.8(2)
N3″-Cd1-N2	103.9(2)	N3**-Cd1-N2	100.9(1)	N3**-Cd1-N2	101.7(2)
N3″-Cd1-N1	86.6(2)	N3**-Cd1-N1	87.9(1)	N3**-Cd1-N1	88.4(2)
Cl1-Cd1-N3″	171.7(2)	Br2*-Cd1-N3**	171.0(1)	I2*-Cd1-N3**	169.5(2)
Cl2-Cd1-N2	153.9(1)	Br1-Cd1-N1	155.2(1)	I1-Cd1-N1	155.8(2)
Cl1′-Cd1-N1	159.9(2)	Br2-Cd1-N2	156.7(1)	I2-Cd1-N2	155.6(2)

^a^ Symmetry codes: (′) 2 − *x*, 2 − *y*, −*z*; (″) 1 − *x*, 2 − *y*, −*z*; (‴) −1 + *x*, *y*, *z*. ^b^ Symmetry codes: (*) 1.5 − *x*, 1.5 − *y*, 1 − *z*; (**) 1 − *x*, 1 − *y*, 1 − *z.*
^c^ Symmetry codes similar to those of complex **2** (see above).

**Table 3 molecules-29-00509-t003:** The to-date structurally characterized metal complexes of neutral dpkoxH with the 2.0111 ^a^ coordination mode ^b^.

Complex	Reference
[Ag^I^_2_(NO_3_)_2_(dpkoxH)_2_]	[32]
[Mn^II^_2_(O_2_CCF_3_)_2_(hfac)_2_(dpkoxH)_2_] ^c^	[33]
[Cu^I^_2_Cl_2_(dpkoxH)_2_]∙2H_2_O	[34]

^a^ Using Harris notation [31]. ^b^ See Figure 6. ^c^ hfac^−^ is the hexafluoroacetylacetonato(−1) ligand.

## Data Availability

Data are contained within the article and Appendix A.

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
