# Peer review of "Reactions of Cadmium(II) Halides and Di-2-Pyridyl Ketone Oxime: One-Dimensional Coordination Polymers"

_molecules, 2024, doi:10.3390/molecules29020509_

Round 1

Reviewer 1 Report

Comments and Suggestions for Authors

Stamou et al. have submitted a manuscript investigating The “Periodic Table” of Di-2-pyridyl Ketone Oxime: Cadmi3 um(II) Complexes.” 

In this work, they reported the employment of dpkoxH in reactions with Cd(II) halides which has resulted in the isolation of three 1D polymeric complexes. Further, The  most important features of this work are: (a) The reactions lead to 1D zigzag polymers with an 1:1 metal to ligand ratio; no monomeric 1:2 complexes could be obtained. (b) The three complexes have interesting molecular and supramolecular structures with the neutral ligand exhibiting a rare coordination mode. (c) The nature of the halogeno ligand  affects slightly the molecular structure and significantly the supramolecular characteristic, with the bromo and iodo complexes being isomorphous and then theirs characterization part.

While the findings can be interesting, they are not presented in a very good or balanced way. In its present state, the overall quality of the work is ok. Upon minor revision it may be suitable for publication. Below are some specific comments I hope can aid the authors in improving the manuscript.

1) Title should be more informative. Even I do not understand what the authors wants to address through this title.  This will increase the impact of this very nice paper.

2) In the synthetic part, ESI-MS for all complexes are missing. The synthetic part should be rewritten in the light of comments.

3) There are too many unnecessary figures related to X-Ray, which makes no sense. Pls move them to supporting information.

Overall my assessment is that this paper deserves publication in Molecules after addressing abovementioned comments and will attract audience interested in such type of materials

Author Response

First of all, we thank Reviewer 1 very much for her/his time to study the ms and the valuable comments which aid us to improve this work. We are glad to see that Reviewer 1 has suggested only minor revisions. All the reviewer’s suggestions were taken into account.

Please find below our point-by-point responses to the reviewer’s revision points/comments/suggestions. For her/his convenience and easy communication, we list the reviewer’s comment (highlighted in green) before our corresponding answer.

While the findings can be interesting, they are not presented in a very good or balanced way. In its present state, the overall quality of the work is ok. Upon minor revision it may be suitable for publication

We thank the reviewer for her/his general comment about the quality of our work.

Title should be more informative. Even I do not understand what the authors wants to address through this title.  This will increase the impact of this very nice paper.

We have changed the title of the ms which is now more informative and addresses better our work.

In the synthetic part, ESI-MS for all complexes are missing. The synthetic part should be rewritten in the light of comments.

The comment is correct. Unfortunately, the three complexes are insoluble in the typical ESI-MS solvents (acetonitrile, methanol, water), so spectra could not be recorded. We even tried dimethylsulfoxide:methanol (1:9 v/v) mixtures, but with no success. We have added a new paragraph in the end of Part 3.2 describing our unsuccessful efforts.

There are too many unnecessary figures related to X-Ray, which makes no sense. Pls move them to supporting information.

We agree with this remark. In accordance with the reviewer’s advice, we have moved Figures 4 and 6 of the originally submitted ms into the “Supplementary Materials” section. Thus Figure 4 has become Figure S2, and Figure 6 is now Figure S4 in the revised version of the ms. The numbering scheme of the other figures (both in the main ms and the “Supplementary Materials” section) has been modified accordingly.

Overall my assessment is that this paper deserves publication in Molecules after addressing abovementioned comments and will attract audience interested in such type of materials。

We thank the reviewer for her/his warm epilogue.

Reviewer 2 Report

Comments and Suggestions for Authors

I have gone through the manuscript and my opinion is that the manuscript needs to undergo minor revision before it can be accepted for publication. My comments are given below. Authors are advised to go through these comments, and respond to them in detail and make necessary changes in the manuscript.

-         The authors should revise and improve the whole abstract by mentioning the causes and rationality alongside the critical overview of this review of this paper.

-         Amendment the key words

-         The introduction section: prefer to compare the estimated results with literatures not start with our research…..

-         I suggest that the authors should provide more comparative samples to give more clarification to the results of the study.

-         The literature in the manuscript should be updated (currently, only a few recently published literature).

-         Line 327: Figure 9. The Raman spectra of complexes 1∙2H2O (black), 2 (blue) and 3 (red).

-         Lines 332-333: This is a clear evidence……not clear

-         Lines 344-345: could you explain that through comparison on the charts

-         J coupling values are missing!

-         Lines 437-438: %H 1.93(calcd) and %H 2.75 (found). Is that acceptable??

-          Establish the mechanism of formation of target compounds.

-         Follow proper instruction from journal policy regarding the reference section.

Author Response

We are grateful to Reviewer 2 for her/his time to evaluate our ms and the valuable comments provided. We are pleased to see that the reviewer recommends minor revision before final acceptance.

Please find below our point-by-point answers to the reviewer’s issues. Most comments were addressed. For her/his convenience and easy communication, we list each comment (highlighted in green) before our answer.

I have gone through the manuscript and my opinion is that the manuscript needs to undergo minor revision before it can be accepted for publication.

We thank the reviewer for her/his general positive comments.

The authors should revise and improve the whole abstract by mentioning the causes and rationality alongside the critical overview of this review of this paper.     

We have revised and improve the “Abstract”. We have added a part in the beginning (“The coordination chemistry……… their zinc(II) analogues”) where we clearly describe the scope of our work, simultaneously providing the readers with the rationale behind our experimental work.

Amendment the key words

We have amended the keywords which are now completely consistent with the subject of the ms.

The introduction section: prefer to compare the estimated results with literatures not start with our research…..

The comment is absolutely correct. We have deleted the first large paragraph of “Introduction” of the originally submitted ms (“Our research efforts……..concerns the latter”) where we had given an overview of our research, which incorporated other research themes from our group. Thus, “Introduction” in the revised ms is now focused only on the general family of ligands, a member of which is the ligand studied in the present work.

I suggest that the authors should provide more comparative samples to give more clarification to the results of the study.

If we understand correctly, this comment is particularly useful. We added two long paragraphs (“One family of such……..pyridyl nitrogen atom” and “The Cd(II)/pyridyl ketoxime chemistry……….their Npyridyl and Noxime atoms”) in “Introduction” of the revised ms. These paragraphs give, according to the suggestion by the reviewer, the existing literature (with all examples) on Cd(II)/pyridyl oxime chemistry; this overview covers both the pyridyl aldoxime and pyridyl ketoxime ligands, and it is not limited to the 2-pyridyl isomers but it is also extended to the 3- and 4-pyridyl ones. Thus, the reader has a clear overview of the relevant literature which heps her/him to understand the significance of the selection of di-2-pyridyl keton oxime as the ligand of study. If the reviewer means with her/his comment that we should incorporate other new compounds in the ms, this would require months of further work. The constant parameters of this study are Cd(II) and the ligand, and the varied parameter (which allows for comparative conclusions) is the nature of the halogeno donor.

The literature in the manuscript should be updated (currently, only a few recently published literature).

We had used the updated literature in the original version of the ms. Few old references (e.g. refs. 31 and 33 of the original version) have been removed, in order to satisfy the reviewer’s suggestion.

Line 327: Figure 9. The Raman spectra of complexes 1∙2H2O (black), (blue) and (red).

The caption of the old Figure 9 (now Figure 7) has been modified, so it is now absolutely clear.

Lines 332-333: This is a clear evidence……not clear

If we understand correctly the comment, we have replaced “This is a clear evidence…..” with “This is an evidence”.

Lines 344-345: could you explain that through comparison on the charts

We are not sure what the reviewer means at this point. Perhaps she/he needs more clarification. Thus, we have added two sentences (“This means ………………dpkoxH compound”) which make the solution situation unambiguous.

 J coupling values are missing!

The J coupling values have been added in Part 3.2 of the “Experimental Section”.

Lines 437-438: %H 1.93(calcd) and %H 2.75 (found). Is that acceptable??

The comment is absolutely correct! This discrepancy can not be justified for a pure sample. We went again through our experimental notebook and we confirmed the found %H value of 2.75, which is unacceptable. Fortunately, we had kept a sample of compound 2 and submitted it to our Microanalytical Service. The new experimental value provided is 2.15%, which is now acceptable.

Establish the mechanism of formation of target compounds.

With no real experimental evidence at hand, we can only speculate about the mechanism of the formation of the complexes. Based on the solid-state structures of the products, we have proposed two mechanisms for the formation of precursor molecules in solution. Thus, we have added a whole paragraph (“A possible scenario………precipitation of the products”) in Section 4 (“Conclusions in Brief and Perspectives”) where we explain the proposed mechanistic ideas.

Follow proper instruction from journal policy regarding the reference section.

We had strictly followed the journal policy regarding the reference section in the originally submitted ms. We have also followed this policy in the small number of the new references in the revised version. Thus, there is no need for changes.

Reviewer 3 Report

Comments and Suggestions for Authors

This paper presents novel research on cadmium coordination complexes with the di-2-pyridyl ketone oxime ligand. While the study provides valuable insights, I recommend major revisions to enhance the clarity, context, and impact of the work prior to publication. Please find below my detailed review highlighting the key areas needing improvement. I believe addressing these concerns will significantly strengthen the manuscript.

Major revisions:

1. The introduction provides good background on oxime and oximato metal complexes, but more context specifically relating to Cd(II)/2-pyridyl ketoxime complexes would strengthen the rationale and importance of this work. The authors mention that Cd(II)/dpkoxH complexes are unknown, but discussing relevant prior Cd(II)/2-pyridyl ketoxime complexes in more detail would better showcase the novelty of the current study.

2. The synthetic methods section needs more experimental detail. Solvent amounts, reaction times and temperatures, crystallization techniques, crystal appearance, etc. should be provided for reproducibility. Characterization methods like IR, Raman, NMR, etc. should be described briefly.

3. The crystal structures and bonding are discussed in depth, but the solution behavior revealed by NMR studies deserves more attention. The proposed decomposition pathways and solution speciation are interesting results that should be expanded on.

4. The conclusion summarizes the key findings but is quite brief. A more comprehensive conclusion recapping the most significant results, comparisons to related systems, and future directions would strengthen the paper.

Overall it is an interesting study and I think addressing these major points will make the manuscript stronger for publication.

Author Response

We are grateful to Reviewer 3 for her/his time to evaluate our ms and the useful comments provided. We are pleased to see that the reviewer recommends acceptance.

Please find below our point-by-point answers to the reviewer’s comments/suggestions. All comments have been addressed. For her/his convenience and easy communication, we list the reviewer’s comment (highlighted in green) before our corresponding answer.

The introduction provides good background on oxime and oximato metal complexes, but more context specifically relating to Cd(II)/2-pyridyl ketoxime complexes would strengthen the rationale and importance of this work. The authors mention that Cd(II)/dpkoxH complexes are unknown, but discussing relevant prior Cd(II)/2-pyridyl ketoxime complexes in more detail would better showcase the novelty of the current study.

The comment, which was also partly raised by Reviewer 2, is absolutely correct. We added two large paragraphs (“One family of such….pyridyl nitrogen atom” and “The Cd(II)/pyridyl ketoxime chemistry……their Npyridyl and Noxime atoms”) in “Introduction” of the revised ms. These paragraphs, according to the suggestion by the reviewer, provide the readers with the existing literature (with all examples) on Cd(II)/pyridyl oxime chemistry; this overview covers both the pyridyl aldoxime and pyridyl ketoxime ligands, and it is not limited to the 2-pyridyl isomers but it is also extended to the 3- and 4-pyridyl ones. Now, the reader has a detailed and clear overview of the relevant literature which helps her/him to understand the importance of the current study.

The synthetic methods section needs more experimental detail. Solvent amounts, reaction times and temperatures, crystallization techniques, crystal appearance, etc. should be provided for reproducibility. Characterization methods like IR, Raman, NMR, etc. should be described briefly.

We are somewhat surprised for this comment. Section 3 (“Experimental Section”), which has three Parts (3.1, 3.2 and 3.3), covers all the experimental details, such as quantities of the reactants, solvent amounts, reaction times, crystallization techniques etc, which are necessary for reproducibility. To show that we respect the reviewer’s suggestion, we have added the crystal appearance in Part 3.2. Also all the instrumentation details (microanalytical service, conductivity measurements, IR, Raman, 1H NMR, 13C NMR, 113Cd NMR) and crystallographic techniques had been briefly described in Parts 3.1 and 3.3, respectively, of the originally submitted ms. It is our common practice to put every experimental detail in the “Experimental Section” (Section 3 in the ms) and not in the “Synthetic Comments” part of “Results and Discussion” (Part 2.1 in the ms). This policy is followed by many journals, including MOLECULES.

The crystal structures and bonding are discussed in depth, but the solution behavior revealed by NMR studies deserves more attention. The proposed decomposition pathways and solution speciation are interesting results that should be expanded on.

The comment is correct. As suggested we have expanded the solution NMR studies in Part 2.3 by giving more explanations and enriching the 113Cd NMR discussion (also incorporating the new references 45 and 46). Thus, the proposed decomposition pathways and solution speciation, represented by equations (4), (5) and (6), are now better supported by the relevant discussion.

The conclusion summarizes the key findings but is quite brief. A more comprehensive conclusion recapping the most significant results, comparisons to related systems, and future directions would strengthen the paper.

We find this comment valuable for the improvement of the quality of our ms. The “Conclusions in Brief and Perspectives” Section (Section 4) of the ms has become much longer in the revised version. Two long paragraphs (“Comparison of the present Cd(II)/dpkoxH complexes………….2.0111 coordination mode (Figure 6)” and “A possible scenario for…….. precipitation of the products”) have been added. The second paragraph addresses also the comment raised by Reviewer 2 and proposes mechanistic ideas about the formation of 1∙2H2O, 2 and 3 (please see our answer to the relevant point by Reviewer 2). The first paragraph covers in detail the revision suggestion by Reviewer 3 for more comprehensive conclusion. Thus, a comparison of the chemistry of our complexes with the chemistry of all the previously reported Cd(II)/2-pyridyl oxime complexes has been added; this comparison is accompanied by the relevant literature references. This paragraph, which is written in a strictly critical manner, complements the two paragraphs (“One family of such……..pyridyl nitrogen atom” and “The Cd(II)/pyridyl ketoxime chemistry…………their Npyridyl and Noxime atoms”) added in “Introduction”, in response to Reviewer 2, which have a purely descriptive character detailing the existing literature on Cd(II)/pyridyl oxime (not confined to 2-pyridyl isomers).

Overall it is an interesting study and I think addressing these major points will make the manuscript stronger for publication.

We thank the reviewer for characterizing our efforts as “an interesting study”. We do believe that we have satisfactorily addressed all her/his revision points and concerns.

Round 2

Reviewer 3 Report

Comments and Suggestions for Authors

After reviewing the revised version I would like to submit this paper for publication in its current form.